# The QPLEX™ Plus Assay Kit for the Early Clinical Diagnosis of Alzheimer’s Disease

**DOI:** 10.3390/ijms241311119

**Published:** 2023-07-05

**Authors:** Hunjong Na, Ki Young Shin, Dokyung Lee, Changsik Yoon, Sun-Ho Han, Jong-Chan Park, Inhee Mook-Jung, Jisung Jang, Sunghoon Kwon

**Affiliations:** 1Department of Electrical and Computer Engineering, Seoul National University, Seoul 08826, Republic of Korea; 2QuantaMatrix Inc., Seoul 08506, Republic of Korea; 3Bio-MAX Institute, Seoul National University, Seoul 08826, Republic of Korea; 4Department of Biochemistry and Biomedical Sciences, College of Medicine, Seoul National University, Seoul 03080, Republic of Korea; 5Neuroscience Research Institute, Medical Research Center, College of Medicine, Seoul National University, Seoul 03080, Republic of Korea; 6SNU Dementia Research Center, College of Medicine, Seoul National University, Seoul 03080, Republic of Korea

**Keywords:** Alzheimer’s disease, dementia, cognition, peripheral blood, algorithm, early diagnosis

## Abstract

We recently developed a multiplex diagnostic kit, QPLEX™ Alz plus assay kit, which captures amyloid-β1-40, galectin-3 binding protein, angiotensin-converting enzyme, and periostin simultaneously using microliters of peripheral blood and utilizes an optimized algorithm for screening Alzheimer’s disease (AD) by correlating with cerebral amyloid deposition. Owing to the demand for early AD detection, we investigate the potential of our kit for the early clinical diagnosis of AD. A total of 1395 participants were recruited, and their blood samples were analyzed with the QPLEX™ kit. The average of QPLEX™ algorithm values in each group increased gradually in the order of the clinical progression continuum of AD: cognitively normal (0.382 ± 0.150), subjective cognitive decline (0.452 ± 0.130), mild cognitive impairment (0.484 ± 0.129), and AD (0.513 ± 0.136). The algorithm values between each group showed statistically significant differences among groups divided by Mini-Mental State Examination and Clinical Dementia Rating. The QPLEX™ algorithm values could be used to distinguish the clinical continuum of AD or cognitive function. Because blood-based diagnosis is more accessible, convenient, and cost- and time-effective than cerebral spinal fluid or positron emission tomography imaging-based diagnosis, the QPLEX™ kit can potentially be used for health checkups and the early clinical diagnosis of AD.

## 1. Introduction

The development of Alzheimer’s disease (AD), the most common form of dementia, is slow and persistent, with a pre-symptom stage lasting over several years to decades [1,2,3,4]. As of 2023, the prevalence of Alzheimer’s dementia among the older population (aged 65 years and older) in the United States is estimated to be approximately 6.7 million individuals [5]. The prominent neuropathological features of AD primarily involve the presence of senile plaques, characterized by the aggregation of amyloid-β (Aβ), and the formation of neuronal neurofibrillary tangles (NFTs) [6]. Generally, AD symptoms begin with mild memory impairment and progress to diverse cognitive impairments, including memory disorder and dysfunctions in complex daily activities [7,8]. The early clinical diagnosis of AD can be defined by means of criteria such as neuropsychological tests [9,10,11]: (a) subjective cognitive decline (SCD) is a continued decline in the self-reported experience in cognitive performance compared to the subject’s previously normal state [12]; and (b) mild cognitive impairment (MCI) is characterized by objective cognitive impairment, including impairment of memory (amnestic) or judgment (non-amnestic) [13]. Both groups are in transitional stages between normal cognition and dementia [14]. The symptoms of SCD are among the early signs of pathological brain aging [15]. Individuals with SCD behavior are associated with Aβ deposition [16]. MCI is also a precursor to AD characterized by neurocognitive dysfunction [17]. Based on estimates for the year 2023, it is projected that approximately 8–11% of the American population aged 65 years and older, corresponding to approximately 5–7 million older individuals, may exhibit MCI [5] and approximately 10–15% of individuals with MCI develop dementia yearly, while 1–2% of unaffected individuals develop dementia [18,19]. These pre-dementia stages could serve as populations for dementia prevention clinical trials [12]. The development of early and accessible diagnostic methods can help to prevent or delay the progression of cognitive deficits and the onset of full-blown AD dementia [20,21]; therefore, targeting the critical precursor steps of SCD or MCI may have a strong potential for the early clinical diagnosis of AD [22].

Increasing efforts to discover biomarkers for AD in the cerebral spinal fluid (CSF) or blood have been ongoing for decades and have led to the discovery of potential biomarkers. CSF analysis revealed amyloid-β1-42 (Aβ42) proteins, Aβ42/Aβ40 ratio, total tau proteins, phosphorylated tau proteins, and neurogranin can be used as a biomarker for AD. In the blood, Aβ42/Aβ40 ratio, amyloid precursor protein (APP)669-711/Aβ42 ratio, tau proteins, and neurofilament light were suggested as biomarkers for AD [23]. Some candidates are related directly to the core pathological features of AD, while others are linked closely to the neurodegeneration of the brain. Compared to CSF-based diagnosis, peripheral blood-based diagnosis has the advantages of reducing the patient’s burden, shortening the inspection time, and lowering the cost of sample collection and examination as a non-invasive method. With these advantages, blood-based diagnosis can be included in health checkups and used for early diagnosis. Although many peripheral biomarkers for AD diagnosis have been reported in recent decades [24,25,26], few have proven to be useful in commercially developed diagnostic kits. Screening peripheral biomarkers for brain diseases has many challenges, such as problems with the sensitivity and specificity of the assay and careful validation work. Such biomarkers can be detectable at relatively low concentrations in the blood because the blood–brain barrier limits the movement of molecules between the central nervous system and the blood vessel system [27]. It is also technically difficult to detect various biomarkers simultaneously within a single assay system [21]. Therefore, it was suggested that multiple combinations of effective peripheral biomarkers with highly sensitive assays might increase diagnostic success for AD [28].

In previous studies, we revealed a novel blood-based biomarker panel consisting of galectin-3 binding protein (LGALS3BP), amyloid-β1-40 (Aβ40), angiotensin-converting enzyme (ACE), and periostin (POSTN) [28,29]. It was demonstrated that LGALS3BP exerts a regulatory role in Aβ production by directly interacting with amyloid precursor protein (APP), consequently impeding APP processing by β-secretase [30]. ACE has been implicated in the processing and metabolism of amyloid β (Aβ) [31]. Therefore, the administration of ACE inhibitors in hypertensive patients diagnosed with AD has been correlated with increased handgrip strength (HGS), preservation of physical capacity, and the prevention of neuromuscular junction (NMJ) degradation [32]. In a recent study, researchers observed a noteworthy relationship between elevated plasma POSTN levels and the progressive decline in physical and cognitive capacities among older adults [33]. We have already produced a bioanalytical platform that can measure four biomarkers with tens of microliters of peripheral blood, the so-called Quantamatrix’s multiplexed diagnostics platform (QPLEX™; Quantamatrix Inc., Seoul, Republic of Korea) Alz plus assay kit. Subsequently, we designed an optimal algorithm, the QPLEX™ algorithm, using the results of the QPLEX™ Alz plus assay kit to predict cerebral amyloid deposition. When the algorithm value exceeds the cutoff value, we assume that the participants may be at risk for AD. The current study shows that a QPLEX™ Alz plus assay kit could be used for cerebral amyloid deposition diagnosis [21,34,35]. However, all studies were developed and applied with the Korean Brain Aging Study for the Early diagnosis and prediction of Alzheimer’s disease (KBASE) cohort. Therefore, applying the kit and the algorithm to other independent cohorts is necessary to verify the performance and bias.

In this paper, our goal is to investigate the potential of the QPLEX™ Alz plus assay kit for the early clinical diagnosis of AD in another independent cohort in South Korea. First, we checked the relationship between the QPLEX™ algorithm values and the four clinically separated groups: cognitively normal individuals (CN), SCD, MCI, and AD. Second, we explored the relationship between our algorithm values and groups divided by the scores of the Mini-Mental State Examination (MMSE) and Clinical Dementia Rating (CDR). The MMSE is a screening tool that provides information about global cognition, and the CDR is a composite evaluation used mainly to determine the presence/absence of functional impairment. Finally, we analyzed the relationship between our algorithm values and the subgroups fractionalized by some factors known to influence the onset of AD, such as sex, age, depression, or apolipoprotein E (ApoE) genotype. The comorbidity of depression is frequently observed in individuals with dementia [36]. As in the case of those three factors, ApoE has been implicated in the pathological changes in AD, including the accumulation of Aβ and tau proteins, which subsequently contribute to neuroinflammation and neuronal injury, ultimately leading to impaired cognitive functions associated with learning and memory [6]. Moreover, we checked the influence of other factors, such as education years, hypertension, diabetes, hyperlipidemia, stroke, angina, thyroid, surgical history, cancer, family history, drinking, smoking, body mass index (BMI), and anxiety. However, at least in our data, there were no significant differences between clinically separated groups (Appendix A), so we did not include them in the in-depth analysis.

## 2. Results

### 2.1. Characteristics of the Participants

This study included 1395 participants (aged 41–92 years) classified into four groups divided by the clinical continuum of AD: 71 CN, 275 SCD, 857 MCI, and 192 AD. The demographic details are shown in Table 1. There were significant differences among the clinically separated groups in terms of age, MMSE score, and CDR score. Moreover, there were significant differences between groups in all the MMSE and CDR test sub-categories.

### 2.2. Demographic Characteristics of the Two Groups Divided by the QPLEX™ Algorithm

In Table 2, QM Alz-N indicates the negative group below the cutoff value and QM Alz-P indicates the positive group equal to or higher than the cutoff value, as classified by the QPLEX™ algorithm. Previous results have shown that a score equal to or higher than the cutoff value indicates a high possibility of cerebral amyloid deposition. Significant differences were demonstrated between QM Alz-N and QM Alz-P in age and CDR (CDR SB, memory, orientation, judgment and problem-solving, community affairs, home and hobbies, and personal care). Except for immediate recall and copying, the scores of other MMSE items (orientation, attention and calculation, memory recall, and language) in QM Alz-N were significantly higher than those in QM Alz-P.

### 2.3. Difference in the QPLEX™ Algorithm Values among Clinically Separated, MMSE-Separated, or CDR-Separated Groups

The average of QPLEX™ algorithm values in each group increased in the order of the clinical progression continuum of AD: CN (0.382 ± 0.150), SCD (0.452 ± 0.130), MCI (0.484 ± 0.129), and AD (0.513 ± 0.136) (Figure 1A). Moreover, the algorithm values between each group showed statistically significant differences (*p* < 0.01). Further, we compared the values of the QPLEX™ algorithm to cognitive evaluation scores of MMSE or CDR (Figure 1B,C). When the groups were divided according to the clinical criteria of each cognitive evaluation, the values of the QPLEX™ algorithm were significantly different among groups in both MMSE and CDR. The MMSE score and algorithm values were analyzed by defining 24 to 30 as normal, 20 to 23 as mild AD dementia, and less than 20 as AD dementia [37]. The algorithm values showed significant differences among the MMSE-separated groups. Furthermore, the CDR score was analyzed by defining 0 as normal, 0.5 as questionable, 1 as mild dementia, and 2 or more as dementia [38]. Similarly, the QPLEX™ algorithm value showed a statistically significant difference among the CDR-separated groups.

### 2.4. Comparison of the QPLEX™ Algorithm Values among the Individual Subgroups Fractionized by Sex, Age, Depression, or ApoE Genotype

We further analyzed whether the clinically separated groups had a significant difference in the QPLEX™ algorithm values even when the participant groups were fractionized by various factors (Figure 2). When the groups were fractionized according to age, sex, or depression, the difference in the algorithm values between the two groups was not significant (Figure 2A,D,G). However, the QPLEX™ algorithm values showed significant differences among the clinically separated subgroups fractionized by age or sex (Figure 2B,C,E,F). In participants without depression, significant differences between subgroups were maintained (Figure 2H). Conversely, in participants with depression, only CN and AD were significantly distinguished, while SCD and MCI were not distinguished (Figure 2I). When the participant groups were fractionated according to ApoE genotyping, the algorithm values of the ApoE ε4-negative group were significantly lower than those of the ApoE ε4-positive group (Figure 2J). In the ApoE ε4-negative group, there was a significant difference in algorithm values among the four subgroups (Figure 2K). However, in the ApoE ε4-positive group, there was no significant difference between the CN subgroup and the other subgroups (Figure 2L). There were only eight CN participants in the ApoE ε4-positive group, which was not a large enough sample size to analyze the statistical significance. However, the algorithm values of the AD subgroup were significantly higher than those of the SCD subgroup in the ApoE ε4-positive group.

### 2.5. ANCOVA Results to Adjust for Covariates, such as Age, Sex, Depression, and ApoE Genotype

The distribution of age, sex, depression, and ApoE genotype differed for each clinical group. When the analysis of covariance (*ANCOVA*) was performed to exclude the effects of these covariates, significant differences were shown between clinical groups regardless of age, sex, or geriatric depression scale (GDS) score (Figure 3A–C). In contrast, when the ApoE genotype was set as a covariate, there was no statistical significance between MCI and AD (Figure 3D). When age, sex, depression, and ApoE genotype were set as covariates simultaneously, there were significant differences between all clinical groups (Figure 3E).

## 3. Discussion

The QPLEX™ Alz plus assay kit adopts a bead-based 3D suspension array system to enhance reactivity and improve sensitivity. As a result, the kit can analyze rare or volume-limited samples, and only 20 µL undiluted human plasma was used per assay in this study. Moreover, the kit is a well-implemented multi-platform capable of measuring four peripheral biomarkers at once using a limited sample and combining them into an algorithm. In previous studies, we identified that these four biomarkers are related to AD [28,29]. Aβ40 is the most representative biomarker of AD. ACE is known as an endopeptidase related to blood pressure control [39,40], but it also acts as an inhibitor of Aβ aggregation [41,42,43]. ACE level and activity were lower in AD patients than in CN individuals [44]. POSTN is related to inflammatory diseases [45,46], and can be found in the cerebral cortex of AD patients [47]. It may be that inflammation is activated and POSTN levels are elevated during AD pathogenesis. LGALS3BP is a receptor for galectin-3, and the binding of LGALS3BP to its ligand inhibits neutrophil activation [48,49].

This study was performed on another independent cohort, including many more participants (n = 1395) compared with the previous studies (n = 300) [21,34]. The target for health checkups are individuals with CN, SCD, or MCI who may potentially develop AD dementia. We included more subjects (n = 1103) with CN, SCD, or MCI than those (n = 236) in the previous studies.

In this paper, we demonstrated that the QPLEX™ Alz plus assay kit could be a useful tool for the early clinical diagnosis of AD. The QPLEX™ algorithm was developed to determine the presence of cerebral amyloid deposition; however, we hypothesized that it could also be used to differentiate between CN and AD based on the fact that, statistically, CN has less amyloid deposition compared to AD. The results suggest that the kit can indeed distinguish the groups according to the clinical progression continuum of AD: CN, SCD, MCI, and AD (Figure 1A). AD neuropathological changes initially target specific brain regions associated with memory, language, and cognitive functions. Consequently, the prodromal symptoms primarily present as impairments in memory, language, and cognitive abilities [5]. Further, there was a significant difference in algorithm values between groups correlating with the score ranges of the MMSE or CDR test (Figure 1B,C) related to memory, language, and cognitive abilities. This implies that the kit can differentiate among groups based on the severity of dementia.

For early diagnosis to be meaningful, it must also be valid for people under the age of 65 years. The correlation between algorithm values and clinical progress was present in patients above 65 years and below 65 years (Figure 2B,C). There were also significant algorithmic value differences among clinical progression in both males and females without bias (Figure 2E,F). This means that the kit can be used regardless of the age and sex of the patient. Notably, a correlation between algorithm values and clinical progression was observed in the absence of depression, but the presence of depression appears to impact the diagnostic accuracy of the QPLEX™ kit. With depression, only CN showed a significant difference with AD (Figure 2I). Although the specific subgroup differences may vary depending on the presence of depression, the QPLEX™ algorithm value proves to be highly valuable in differentiating between CN and AD, regardless of the presence or absence of depression. Further research on our kit is necessary to incorporate depression and effectively distinguish various clinical symptoms. Conversely, the presence or absence of the ApoE gene shows a significant difference in the QPLEX™ algorithm value (Figure 2J), because people with the ApoE gene are more likely to develop MCI or AD [50,51]. Additional research on the QPLEX™ kit will be necessary to differentiate various clinical symptoms while incorporating ApoE results in the future.

We also performed *ANCOVA* to adjust for covariates, such as age, sex, depression, and ApoE genotype (Figure 3). Although the distribution of age, sex, and depression differed between cognitive states, the *ANCOVA* results adjusted for these covariates showed statistically significant differences among all clinical groups (Figure 3A–C). These also indicate that the QPLEX™ kit can distinguish the cognitive states regardless of age, sex, and depression. In *ANCOVA* with ApoE genotype set as a covariate, significant differences were present among all groups except between the MCI and AD groups (Figure 3D). This lack of significance between MCI and AD can be attributed to the fact that high ApoE positivity was considered, but differences in the other factors were not adjusted for. When age, sex, depression, and ApoE genotype were all set as covariates, the QPLEX™ algorithm values showed significant differences among all groups (Figure 3E). The *ANCOVA* results confirm that the QPLEX™ algorithm values are related to cognitive states.

One intended application of the kit is for use in routine screening for AD among the general population. Our kit has four main advantages that make it highly suitable for this purpose: Firstly, it detects multiple biomarkers, which has been shown to increase the accuracy of diagnosing AD [28], predicting the conversion from MCI to AD [52,53], identifying MCI patients susceptible to AD [54], and predicting cerebral Aβ deposition [24,26,28]. Secondly, it detects these biomarkers in blood samples, which are easier to acquire and more cost- and time-effective than detecting biomarkers in CSF [25,55,56,57,58,59]. Thirdly, it requires only a small sample volume of 20 µL undiluted plasma per analysis. Fourthly, the kit utilizes highly stable magnetic beads that are amenable to automation [21].

Additionally, as in previous studies, participants were analyzed by dividing them into negative and positive, based on the algorithm cutoff value (Table 2). The *t*-test results show a statistically significant difference between QM Alz-N and QM Alz-P in all items of MMSE and CDR except for immediate recall and copying of MMSE. However, the magnitude of the difference seems small. This can be explained by the characteristics of the PREMIER cohort, and we used the CDR score as an example. The purpose of the consortium is early diagnosis, especially focusing on changes in blood biomarkers, genetic data, and pathological data during the progression from MCI to AD with a longitudinal study; hence, more than half of the recruited participants had MCI (Table 1). The proportion of SCD and MCI participants exceeded 80% of the total cohort, with 93% of participants in these categories having a CDR score of 0.5. As a result, the CDR scores converged to 0.5 for both QM Alz-N and QM Alz-P. In the KBASE cohort, CDR scores differed according to the clinical state, but there was no difference according to amyloid deposition [21]. When the KBASE cohort was classified by PET results, there seemed to be a difference in CDR scores between the two groups, but this is a result of the difference in the CN/MCI/dementia ratio [34].

The study also had some limitations. First, proteomics screening and statistical analysis show that blood biomarkers LGALS3BP, ACE, and POSTN effectively screen for cerebral amyloid depositions and clinically diagnose Alzheimer’s disease, but the theoretical basis is still lacking. Additional research will be needed to supplement these theoretical grounds. Second, a longitudinal study will be needed to observe the change in our assay depending on the disease progression. Third, owing to limitations in accessibility, the study was conducted only for the Korean cohort. Analysis of cohorts of different regions and races is needed to confirm the possibility of universal applicability. Fourth, various types of dementia or MCI need to be compared using our assay to investigate the possibility of distinguishing these from one another. Fifth, there were only eight participants in the ApoE ε4-positive CN group, which was an insufficient sample size for statistical significance analysis. ApoE positivity is distributed with a lower probability in the CN group, and the recruitment rate for the CN group was low due to the characteristics of the cohort, resulting in a shortage of participants in the ApoE ε4-positive CN group. Sixth, only cognition tests, such as MMSE and CDR, were utilized in this study. However, there is a need for additional validation of the relationship between the QPLEX™ algorithm and other more sensitive and specific tests, such as the Montreal Cognitive Assessment (MoCA). Seventh, we only conducted the prediction of cerebral amyloid deposition using the QPLEX™ algorithm. However, since tauopathy is highly relevant in the pathophysiology and progression of AD, future research and validation will also be necessary using tau PET.

In conclusion, the QPLEX™ Alz plus assay kit, a multiplex system to analyze four blood biomarkers consisting of LGALS3BP, Aβ40, ACE, and POSTN simultaneously, showed potential as a screening tool for AD. In particular, our kit could be a useful detection tool for the early clinical diagnosis of AD, i.e., for SCD or MCI. Our kit could be a helpful diagnostic tool for cognitive impairments at health checkups because the kit can measure multiple blood biomarkers using only tens of microliters of blood.

## 4. Materials and Methods

### 4.1. Participants

In total, 1633 participants were included from 14 referral hospitals in the Republic of Korea. Most of the participants were recruited from the Samsung Medical Center (n = 579), the Soonchunhyang University Bucheon Hospital (n = 313), and the Kangwon National University Hospital (n = 265). The rest (n = 476) were from 11 hospitals across Korea, including the Korea University Guro Hospital (n = 56). These participants were recruited from a nationally funded, nationwide multicenter study named Precision medicine platform for mild cognitive impairment, based on Multi-omics, Imaging, Evidence-based R&BD (PREMIER) consortium (conducted between May 2019 and December 2022) in South Korea [60]. The aim of this consortium is to establish a platform for the development of the early diagnosis and precision medicine-based treatment of dementia by enrolling participants with various cognitive states, developing blood-based biomarkers, generating genetic data, and developing imaging-data- and clinical-data-based algorithms as a diagnostic or predictive tool. According to the purpose of the consortium, the cohort recruited mainly MCI participants at the pre-stage of dementia.

### 4.2. Clinical Diagnosis

An experienced neurologist diagnosed participants with SCD, MCI, or AD dementia according to relevant diagnostic criteria [61]. Based on the recommendation of Molinuevo et al. [62], the criteria for SCD are as follows: (1) self-experienced persistent decline in cognitive performance compared to previously normal state, (2) normal performance on all neuropsychological tests, and (3) cannot be explained by other psychiatric or neurologic diseases. Based on Petersen’s criteria [63], the criteria for MCI are as follows: (1) cognitive complaint, preferably corroborated by an informant; (2) objective cognitive impairment for age and educational level; (3) relatively preserved general cognition; (4) intact activities of daily living; and (5) not demented. The criteria for AD dementia are based on the proposal by the National Institute on Aging—Alzheimer’s Association (NIA–AA) Research Framework [64]. AD patients exhibited CDR scores ranging from 0.5 to 3, and their MMSE scores were 10 or higher, indicating their suitability for Seoul neuropsychological screening battery-dementia version (SNSB-D) testing [65].

### 4.3. Cognition Tests

All participants underwent MMSE and CDR (Table 1). The MMSE consists of tests for orientation, immediate recall, attention and calculation, memory recall, language, and copying. The CDR consists of tests for memory, orientation, judgment and problem-solving, community affairs, home and hobbies, and personal care. The details and protocols of neuropsychological assessment are described in a previous report [66]. The diverse interpretation of the MMSE scores and their relationship with the severity of dementia were depicted as several steps [3,67,68,69]. Using one of the ranging methods of MMSE, we split the participants into three groups: normal, with a score range from 24 to 30; mild dementia, with a score range from 20 to 23; and moderate to severe dementia, with a score range below 19 [37]. The CDR scale is a global clinical staging method for AD [70]. Using a simplified description of the AD stages according to the CDR score, we also split CDR groups as follows: 0 = no dementia or normal; 0.5 = questionable; 1 = mild; and 2 to 3 = moderate to severe [38].

### 4.4. Short Geriatric Depression Scale—Korean Version (SGDS-K)

The SGDS-K, which stands for the Korean version of the Elderly Depression Scale [71], was derived from the Geriatric Depression Scale (GDS) [72]. It comprises 15 yes/no questions, resulting in a total score of 15 points. The SGDS-K evaluates symptoms of depression experienced during the previous week [73,74]. Using one of the ranging methods of GDS, we split the participants into two groups: normal, with a score range from 0 to 4, and depression, with a score range of 5 or higher [75,76].

### 4.5. Blood Sampling and Storage

Whole-blood samples were collected in K2 EDTA tubes (BD Vacutainer Systems, Plymouth, UK). The blood sample tubes were centrifuged at 700× *g* for 5 min at room temperature, and plasma supernatants were stored at −80 °C [21,34].

### 4.6. Exclusion Criteria of the Participants

Figure 4 shows the criteria and number for the inclusion and exclusion of participants. A total of 1633 participants were recruited. Among them, those without blood samples (n = 10), ApoE genotype (n = 19), MMSE (n = 57), CDR (n = 70) tests missing, or with other forms of disease (N = 82) were excluded from the analysis. Finally, QPLEX™ assay and data analysis were performed on 1395 participants.

### 4.7. QPLEX™ Alz Plus Assay

QPLEX™ kit is a bead-based 3D suspension array system for multiplex analysis in a single well [77,78]. The micro-sized beads, referred to as microdisks, are graphically encoded using photolithography. By reacting multiple microdisks pre-coupled with specific markers according to their specific codes, and analyzing the fluorescence signal for each code, multiple markers can be analyzed simultaneously in one well. Briefly, 35 µL diluted human plasma samples and 35 µL biotin-conjugated detection antibodies were incubated with microdisks in a 96-well plate for 90 min at room temperature with shaking at 1000 rpm. The reacted microdisks were washed with 0.1% BSA buffer and incubated with 50 µL of 2 µL/mL R-phycoerythrin-conjugated streptavidin for 15 min at room temperature with shaking at 1000 rpm. After three washes, the microdisks were re-suspended in 100 µL of 0.1% BSA buffer and analyzed with Quantamatrix’s multiplex assay platform (QMAP™).

The QPLEX™ algorithm was developed to diagnose cerebral amyloid deposition. This algorithm was developed through logistic regression with the quantitative values of LGALS3BP, Aβ40, ACE, and POSTN obtained from the QPLEX™ Alz plus assay kit as the independent variables and the results of PET imaging performed at the Seoul National University Hospital as the dependent variables [21,34]. Subsequently, the cutoff value, sensitivity, and specificity were obtained through receiver operating characteristic (ROC) curve analysis. The equation of the QPLEX™ algorithm was as follows:$$P_{i}=\frac{E}{1+E}$$
$$E=exp(a_{1}\times LGALS3BP+a_{2}\times A\beta 40+a_{3}\times ACE+a_{4}\times POSTN+C)$$

($P_{i}$, predicted probabilities; $a_{n}$, coefficient values for each biomarker, with $a_{1}$ = −0.00066, $a_{2}$ = 0.008, $a_{3}$ = −0.00662, and $a_{4}$ = 0.13224; and *C* = 1.24777, which is a constant. The quantitative values of each biomarker obtained with the QPLEX™ Alz plus kit were multiplied by the coefficient values, and $P_{i}$ was calculated).

The cutoff value to maximize sensitivity and specificity for screening cerebral amyloid deposition determined by ROC curve analysis was 0.461.

### 4.8. Data Analysis

All statistical analyses were performed using Medcalc 20.115 (Ostend, Belgium), and the figures were generated using GraphPad Prism 5 (San Diego, CA, USA). The comparison between groups was performed with an independent t-test, analysis of variance (*ANOVA*) with a Student–Newman–Keuls *post hoc* test, or *ANCOVA*. *p* < 0.05 was considered statistically significant. In *ANCOVA*, the *p*-value was Bonferroni-corrected. Figure 5 illustrates the overall data analysis process, showing the criteria used to create subgroups and the number of participants in each subgroup.

## Figures and Tables

**Figure 1 ijms-24-11119-f001:**
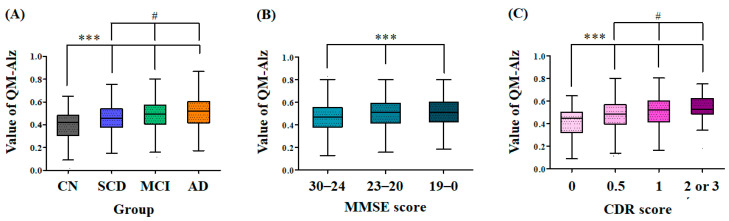
Difference of the QPLEX™ algorithm values among the clinically separated groups, MMSE-separated, or CDR-separated groups. The values of the QPLEX™ algorithm showed statistically significant differences among the (**A**) four clinically separated groups (CN (n = 71), SCD (n = 275), MCI (n = 857), and AD (n = 192)), (**B**) MMSE-separated groups (30–24 (n = 923), 23–20 (n = 265), and 19–0 (n = 207)), and (**C**) CDR-separated groups (CDR 0 (n = 98), CDR 0.5 (n = 1153), CDR 1 (n = 117), and CDR 2 (n = 27)). Data represent mean ± standard deviation. *** *p* < 0.01 compared with the (**A**) CN group, (**B**) the group ranging from 24 to 30 in MMSE scores, and (**C**) the group scoring 0 in CDR, one-way *ANOVA* with Student–Newman–Keuls *post hoc* comparison, respectively; ^#^ *p* < 0.05 compared with the (**A**) SCD group, and (**C**) the group scoring 0.5 in CDR, one-way *ANOVA* with Student–Newman–Keuls *post hoc* comparison, respectively; CN, cognitively normal; SCD, subjective cognitive decline; MCI, mild cognitive impairment; AD, Alzheimer’s disease; MMSE, Mini-Mental State Examination; CDR, Clinical Dementia Rating.

**Figure 2 ijms-24-11119-f002:**
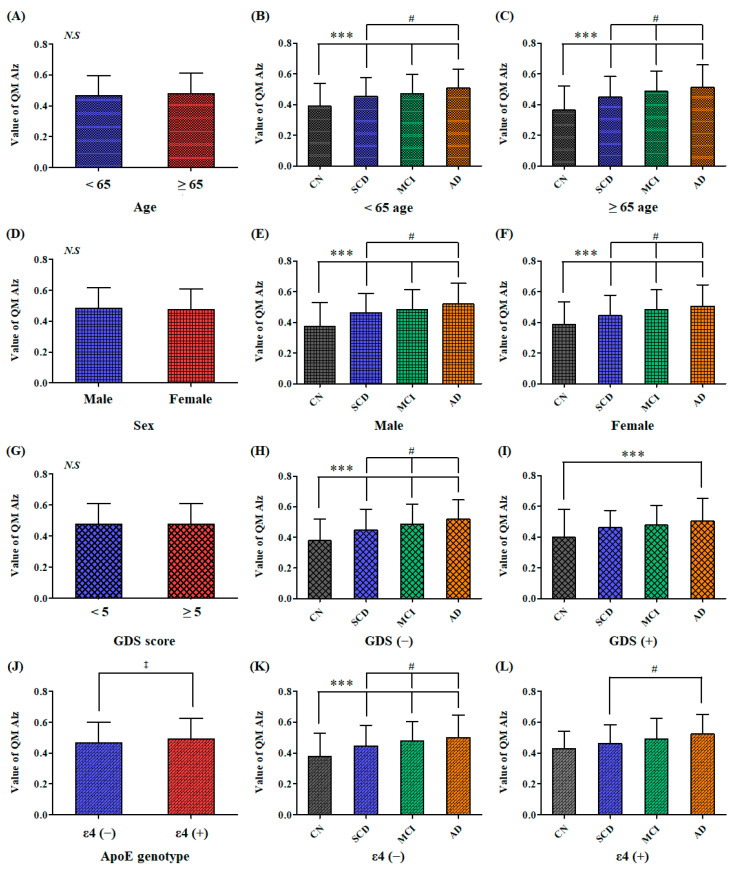
Difference of the QPLEX™ algorithm values within the subdivided groups by various factors. (**A**) Comparison between the groups under 65 and over 65. (**B**,**C**) Comparison among the clinically separated subgroups by age. (**D**) Comparison between the male and female groups. (**E**,**F**) Comparison among the clinically separated subgroups by sex. (**G**) Comparison between the group without depression and the group with depression. (**H**,**I**) Comparison among the clinically separated subgroups by depression. (**J**) Comparison between ApoE ε4-negative (ε4 (−)) and ApoE ε4-positive (ε4 (+)) group. (**K**,**L**) Comparison among the clinically separated subgroups by ApoE genotype. Data represent mean ± standard deviation; *N.S.*, no significance; ^‡^ *p* < 0.01 compared with the ε4 (−) group, an independent t-test; *** *p* < 0.05 compared with the CN group, one-way *ANOVA* with Student–Newman–Keuls *post hoc* comparison; ^#^ *p* < 0.05 compared with the SCD group, one-way *ANOVA* with Student–Newman–Keuls *post hoc* comparison; CN, cognitively normal; SCD, subjective cognitive decline; MCI, mild cognitive impairment; AD, Alzheimer’s disease; GDS, Geriatric Depression Scale; ApoE, Apolipoprotein E.

**Figure 3 ijms-24-11119-f003:**
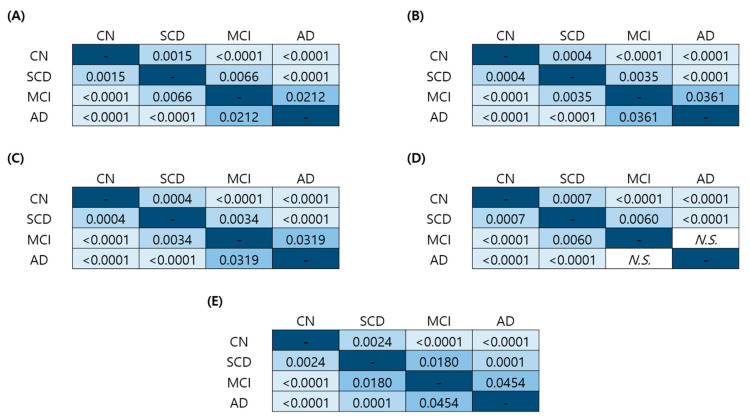
Bonferroni-corrected *p*-values of *ANCOVA*. *p*-values of *ANCOVA* when covariates were set as (**A**) age, (**B**) sex, (**C**) GDS score, (**D**) ApoE genotype, and (**E**) all four factors combined. *ANCOVA*, analysis of covariance; *N.S.*, not significant; GDS, Geriatric Depression Scale; ApoE, Apolipoprotein E.

**Figure 4 ijms-24-11119-f004:**
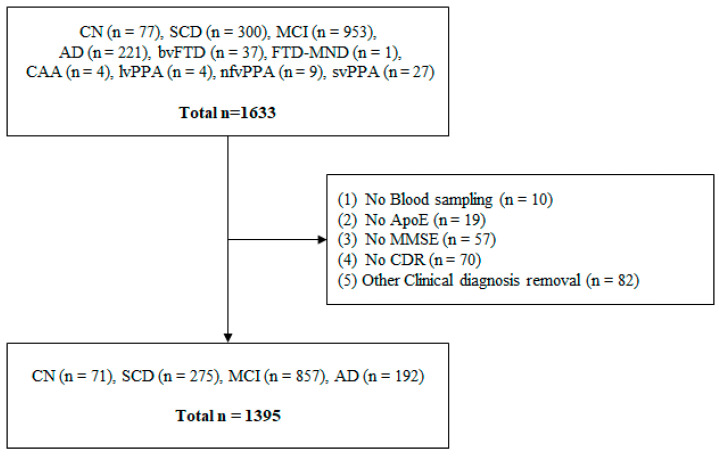
Flowchart for participant selection. CN, cognitively normal; SCD, subjective cognitive decline; MCI, mild cognitive impairment; AD, Alzheimer’s disease; bvFTD, behavioral variant frontotemporal dementia; FTD-MND, frontotemporal dementia with motor neuron disease; CAA, cerebral amyloid angiopathy; lvPPA, logopenic variant primary progressive aphasia; nfvPPA, nonfluent/agrammatic variant primary progressive aphasia; svPPA, semantic variant primary progressive aphasia; ApoE, apolipoprotein E; MMSE, Mini-Mental State Examination; CDR, clinical dementing rating.

**Figure 5 ijms-24-11119-f005:**
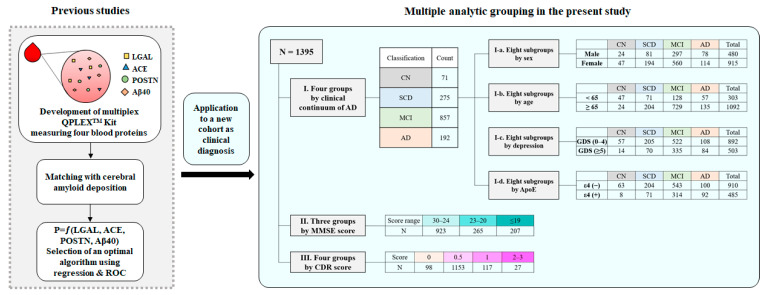
Overall data analysis process flow. In previous studies, we developed the QPLEX™ kit to quantify four blood biomarkers (LGAL, ACE, POSTN, and Aβ40) and indicated its optimal algorithm to screen cerebral amyloid deposition. In this study, we applied the kit and algorithm to a new additional cohort for the clinical diagnosis of AD. A total of 1395 participants were divided according to the clinical continuum of AD, MMSE score, and CDR score, and the differences in algorithm values among groups were compared. The clinical continuum of AD was further analyzed by subdividing according to sex, age, and ApoE genotype. According to the clinical criteria, the MMSE scores were grouped into normal from 24 to 30, mild AD dementia from 20 to 23, and AD dementia for scores lower than 20. Similarly, the CDR scores were grouped into normal for 0, questionable for 0.5, mild dementia for 1, and dementia for two or more. LGAL, galectin-3 binding protein; ACE, angiotensin-converting enzyme; POSTN, periostin; Aβ40, amyloid-β1-40; ROC, receiver operating characteristic; CN, cognitively normal; SCD, subjective cognitive decline; MCI, mild cognitive impairment; AD, Alzheimer’s disease; ApoE, Apolipoprotein E; MMSE, Mini-Mental State Examination; CDR, Clinical Dementia Rating.

**Table 1 ijms-24-11119-t001:** Demographic data of the participants (N = 1395).

	CN	SCD	MCI	AD
Number	71 (5%)	275 (20%)	857 (61%)	192 (14%)
Age	61.49 ± 9.44	69.79 ± 7.80 ^a^	72.86 ± 8.15 ^a^	70.26 ± 9.44 ^a^
Sex (M/F)	24/47	81/194	297/560	78/114
ApoE genotype (ε4 +/−)	8/63	71/204	314/543	92/100
MMSE total	27.93 ± 2.24	27.92 ± 1.91	24.31 ± 4.06 ^a^	19.33 ± 4.73 ^a^
Orientation	9.85 ± 0.36	9.84 ± 0.43	8.59 ± 1.69 ^a^	5.94 ± 2.07 ^a^
Immediate recall	2.96 ± 0.20	2.96 ± 0.22	2.90 ± 0.37	2.81 ± 0.49 ^a^
Attention and Calculation	4.21 ± 1.12	4.17 ± 1.03	3.32 ± 1.51 ^a^	2.36 ± 1.75 ^a^
Memory recall	2.37 ± 0.76	2.27 ± 0.96	1.35 ± 1.11 ^a^	0.66 ± 0.99 ^a^
Language	7.59 ± 0.77	7.74 ± 0.56	7.32 ± 1.01 ^a^	6.92 ± 1.34 ^a^
Copying	0.94 ± 0.23	0.94 ± 0.24	0.81 ± 0.39 ^a^	0.65 ± 0.48 ^a^
CDR score	0.18 ± 0.24	0.46 ± 0.14 ^a^	0.50 ± 0.16 ^a^	0.94 ± 0.49 ^a^
CDR SB	0.21 ± 0.30	0.67 ± 0.48 ^a^	1.63 ± 1.35 ^a^	5.65 ± 2.87 ^a^
Memory	0.18 ± 0.24	0.46 ± 0.15 ^a^	0.60 ± 0.28 ^a^	1.24 ± 0.57 ^a^
Orientation	0.00 ± 0.00	0.05 ± 0.18	0.32 ± 0.36 ^a^	1.07 ± 0.55 ^a^
Judgment and Problem-solving	0.01 ± 0.06	0.07 ± 0.18	0.28 ± 0.33 ^a^	0.95 ± 0.53 ^a^
Community Affairs	0.00 ± 0.00	0.03 ± 0.12	0.19 ± 0.32 ^a^	0.97 ± 0.61 ^a^
Home and Hobbies	0.02 ± 0.10	0.06 ± 0.17	0.20 ± 0.32 ^a^	0.95 ± 0.61 ^a^
Personal care	0.00 ± 0.00	0.01 ± 0.05	0.04 ± 0.23	0.47 ± 0.68 ^a^

Data represent mean ± standard deviation; ^a^, *p* < 0.05 compared with the CN group, one-way *ANOVA* with Student–Newman–Keuls *post hoc* comparison; CN, cognitively normal; SCD, subjective cognitive decline; MCI, mild cognitive impairment; AD, Alzheimer’s disease; ApoE, Apolipoprotein E; MMSE, Mini-Mental State Examination; CDR, Clinical Dementia Rating; CDR SB, Clinical Dementia Rating Sum of Boxes.

**Table 2 ijms-24-11119-t002:** Demographic characteristics of the two groups divided by the QPLEX™ algorithm.

	QM Alz-N	QM Alz-P	*p*-Value
Number	595	800	
Age	70.71 ± 9.04	71.76 ± 8.49	*p* = 0.0283
Sex (M/F)	212/383	268/532	
ApoE genotype (ε4 +/−)	184/411	301/499	
MMSE total	25.15 ± 4.28	24.05 ± 4.70	*p* < 0.0001
Orientation	8.80 ± 1.75	8.34 ± 2.03	*p* < 0.0001
Immediate recall	2.92 ± 0.34	2.90 ± 0.38	*N.S.*
Attention and Calculation	3.53 ± 1.50	3.31 ± 1.58	*p* = 0.0068
Memory recall	1.62 ± 1.13	1.39 ± 1.18	*p* = 0.0002
Language	7.43 ± 0.94	7.31 ± 1.07	*p* = 0.0289
Copying	0.84 ± 0.36	0.81 ± 0.40	*N.S.*
CDR score	0.50 ± 0.24	0.57 ± 0.32	*p* < 0.0001
CDR SB	1.60 ± 1.82	2.16 ± 2.38	*p* < 0.0001
Memory	0.58 ± 0.37	0.68 ± 0.43	*p* < 0.0001
Orientation	0.29 ± 0.43	0.40 ± 0.49	*p* < 0.0001
Judgment and Problem-solving	0.25 ± 0.38	0.36 ± 0.46	*p* < 0.0001
Community Affairs	0.20 ± 0.38	0.30 ± 0.49	*p* < 0.0001
Home and Hobbies	0.21 ± 0.37	0.31 ± 0.49	*p* < 0.0001
Personal care	0.06 ± 0.27	0.11 ± 0.39	*p* = 0.0059

Data represent mean ± standard deviation; QM Alz-N, the negative group that has below-cutoff values in the QPLEX™ algorithm, resulting from the QPLEX™ kit; QM Alz-P, the positive group that has values equal to or over the cutoff values in the QPLEX™ algorithm, resulting from the QPLEX™ kit; *N.S.*, not significant; ApoE, Apolipoprotein E; MMSE, Mini-Mental State Examination; CDR, Clinical Dementia Rating; CDR SB, Clinical Dementia Rating Sum of Boxes.

## Data Availability

The data presented in this study are available in the insert article.

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
