# Peer review of "The QPLEX™ Plus Assay Kit for the Early Clinical Diagnosis of Alzheimer’s Disease"

_ijms, 2023, doi:10.3390/ijms241311119_

Round 1

Reviewer 1 Report

The article by Hunjong Na et al. aimed to investigate the potential of the QPLEX™ Alz plus assay kit for early clinical diagnosis of AD. Previously this study had been carried out in other cohorts of patients in Korea, however, the “novelty” of this paper was applying the kit and the algorithm to other independent cohorts is necessary to verify the performance and bias. I consider that this manuscript lacks novelty, has serious flaws, additional experiments are needed and research is not conducted correctly.

Major observations:

• Abstract

1. The 4 proteins detected by the multiplex diagnostic kit have not been specified.

2. Some abbreviations need to be defined.

Introduction

3. It is important that the authors briefly describe the generalities of galectin-3 binding protein (LGALS3BP), angiotensin-converting enzyme (ACE), and periostin (POSTN) and the rationale for using these molecules as biomarkers in AD.

4. Add generalities and briefly pathophysiology of AD

5. Include the relevance of APOE as a predisposing factor for AD.

• Results

6. There is almost a decade difference between the CN group compared to the other groups. So is it valid to compare the variables studied?

7. From a statistical point of view, is it valid to compare such unbalanced populations (for example CN= 71 and MCI= 857)?

8. Specify in the text or Figure caption the number of participants representing each group.

• Methods:

9. In addition to the scales used MMSE and CDR to determine global cognition and functional impairment, what were the inclusion and exclusion criteria of the participants? At what stage of the disease were the patients diagnosed with AD?

10. Because there are multiple risk factors for dementia such as less education, hypertension, hearing impairment, smoking, obesity, depression, physical inactivity, diabetes, and infrequent social contact (10.1016/S0140-6736(20)30367-6) it is important that authors consider some of these variables as well. This analysis would bring novelty to the manuscript.

11. Since this new cohort is being performed to validate its algorithm, including a group with PET results would be necessary.

12. Justify the score established for the MMSE, not those used in other studies.

13. Since the criteria for the diagnosis of subjective cognitive impairment and moderate cognitive impairment are being specified, the criteria for the diagnosis of AD should also be specified. Why did you use the NINDS-ADRDA criteria, when there are more current criteria (NIA-AA; 10.1016/j.jalz.2018.02.018)?

•            Discussion

14. The authors should discuss the limitation of only considering the prediction of cerebral amyloid deposition in the QPLEX™ algorithm, given that tauopathy is highly relevant in the pathophysiology and clinical progression of AD.

15. Discuss the limitation of using MMSE and CDR and not other more sensitive and specific ones such as “The Montreal Cognitive Assessment, MoCA” (10.1111/j.1532-5415.2005.53221.x)

16. Mention whether other work groups have also identified these four molecules as biomarkers for AD.

•            References

17. Around 50% of the references are from 2016 or earlier. Update as much as possible.

Minor observations

1. Review lines 58 to 62 as there are repetitive elements.

2. Improve the quality of the figures.

3. It remains to cite Figure 4 in the text.

4. Homogenize abbreviations throughout the text.

Minor editing of the English language required

Author Response

Q1) The 4 proteins detected by the multiplex diagnostic kit have not been specified.
A1) Following your advice, we specified four proteins.

Q2) Some abbreviations need to be defined.
A2) We added the definition of abbreviations.

Q3) It is important that the authors briefly describe the generalities of galectin-3 binding protein (LGALS3BP), angiotensin-converting enzyme (ACE), and periostin (POSTN) and the rationale for using these molecules as biomarkers in AD.
A3) We added generalities of each protein to our previous research part (page 2, lines 88-98).

Q4) Add generalities and briefly pathophysiology of AD
A4) We added this content at the beginning of the introduction (page 1, lines 44-46).

Q5) Include the relevance of APOE as a predisposing factor for AD.
A5) We added this content at the end of the introduction (page 3, lines 121-125).

Q6) There is almost a decade difference between the CN group compared to the other groups. So is it valid to compare the variables studied?
A6) We are aware that the average age differs for each clinical group. To adjust for this age difference, we performed an ANCOVA analysis with age as a covariate, and showed that there is a significant difference in algorithm values between groups regardless of age (Figure 3A).

Q7) From a statistical point of view, is it valid to compare such unbalanced populations (for example CN= 71 and MCI= 857)?
A7) This study is part of the PREMIER consortium, a nationally-funded nationwide multicenter study for the purpose of early diagnosis, especially focusing on intermediate state changes in blood biomarkers, genetic data, and pathological data during the progression from MCI to AD with a longitudinal study; hence, more than half of the recruited participants had MCI (page 9, lines 315-318).
Below are the estimated marginal means for each group, with no overlap between groups in the 95% CI. This means that even CN, which has the smallest population, has a statistically significant population.

N

Mean

Std. Error

95% Confidence interval

CN

71

0.3819

0.0156

0.3513

0.4124

SCD

275

0.4519

0.0079

0.4364

0.4675

MCI

857

0.4837

0.0045

0.4749

0.4925

AD

192

0.5130

0.0095

0.4945

0.5316

Q8) Specify in the text or Figure caption the number of participants representing each group.
A8) The number of participants for each group was detailed in Figure 5 (= Figure 4 in the previous version). However, it seems that the figure was too reduced to fit the format and was not clearly visible. Therefore, Figure 5 has been enlarged. Also, as you commented, the number of participants for each group has been specified in the caption of Figure 1.

Q9) In addition to the scales used MMSE and CDR to determine global cognition and functional impairment, what were the inclusion and exclusion criteria of the participants? At what stage of the disease were the patients diagnosed with AD?
A8) We added the inclusion and exclusion criteria of the participants in the method section (page 11, lines 415-428). To illustrate this, we added Figure 4 in the method section. We also specified the criteria for the diagnosis of AD (page 11, lines 381-385).

Q10) Because there are multiple risk factors for dementia such as less education, hypertension, hearing impairment, smoking, obesity, depression, physical inactivity, diabetes, and infrequent social contact (10.1016/S0140-6736(20)30367-6) it is important that authors consider some of these variables as well. This analysis would bring novelty to the manuscript.
A10) Following your advice, we checked for significant differences between clinical groups for factors such as education, high blood pressure, diabetes, hyperlipidemia, stroke, angina, thyroid, surgical history, cancer, family history, alcohol consumption, smoking, BMI, depression, and anxiety. As a result, we found a significant relationship between depression and clinical groups. We added content on the diagnosis protocol of depression (page 11, lines 402-408), results (page 6, lines 190-238), and discussion (page 9, lines 277-284).

Q11) Since this new cohort is being performed to validate its algorithm, including a group with PET results would be necessary.
A11) We agree with your opinion. However, due to the cost and time issue of PET examination, we were unable to perform PET examination on over 1,000 participants. Therefore, we focused on clinical diagnosis in this paper.

Q12) Justify the score established for the MMSE, not those used in other studies.
A12) Rather than arbitrarily dividing the score, we thought that dividing the score using a validated method in previous research is a more reliable method. As mentioned in the Method section (page 11, lines 392-397), we chose the most appropriate way to divide our cohort among several ways to divide MMSE scores. Below figure shows the classification criteria of MMSE from the reference we chose. When we divided participants into 10-19 and 0-9, the difference in the number of participants between each group was too large, so we revised the criteria to 0-19.

Score

Interpretation

24-30

Normal range

20-23

Mild cognitive impairment or possible early-stage/
mild Alzheimer’s disease

10-19

Middle-stage/moderate Alzheimer’s disease

0-9

Late-stage/severe Alzheimer’s disease

Q13) Since the criteria for the diagnosis of subjective cognitive impairment and moderate cognitive impairment are being specified, the criteria for the diagnosis of AD should also be specified. Why did you use the NINDS-ADRDA criteria, when there are more current criteria (NIA-AA; 10.1016/j.jalz.2018.02.018)?
A13) According to your recommendation, we updated the method section with NIA-AA criteria (page 11, lines 381-385). We confirmed that our AD diagnosis not only fulfill the NINDS-ADRDA criteria but also meets the NIA-AA criteria.

Q14) The authors should discuss the limitation of only considering the prediction of cerebral amyloid deposition in the QPLEX™ algorithm, given that tauopathy is highly relevant in the pathophysiology and clinical progression of AD.
A14) We added this content at the limitations section (page 10, lines 343-346).

Q15) Discuss the limitation of using MMSE and CDR and not other more sensitive and specific ones such as “The Montreal Cognitive Assessment, MoCA” (10.1111/j.1532-5415.2005.53221.x)
A15) We added this content at the limitations section (page 10, lines 339-343).

Q16) Mention whether other work groups have also identified these four molecules as biomarkers for AD.
A16) We wrote about research results that studied the relationship between each protein and AD in the introduction section (page 2, lines 90-98) and discussion section (page 8, lines 247-253). There was no group that used these four molecules simultaneously. PMID: 33407839, PMID: 28330509, and PMID: 31605717 are researches that our group also participated in.

Q17) Around 50% of the references are from 2016 or earlier. Update as much as possible.
A17) Following your recommendation, we updated references to recent studies.

Year

References

~2000

8

~2005

6

~2010

3

~2015

15

~2020

27

~2023

19

Minor observations

Q1) Review lines 58 to 62 as there are repetitive elements.
A1) The sentence was too long and there was a problem in conveying intent. We revised the sentence to make it clearer in meaning (page 2, lines 68-71).

Q2) Improve the quality of the figures.
A2) To increase visibility, we adjusted the text size of each figure. Figure 5 has been modified to a vertical version.

Q3) It remains to cite Figure 4 in the text.
A3) As per your advice, we cited all figures in the text.

Q4) Homogenize abbreviations throughout the text.
A4) We checked abbreviations throughout the text.
QPLEX™ is a kit name and QMAP™ is an analysis equipment name.

The attached manuscript reflects all the comments and suggestions of the reviewers.
For the quality of English, we commissioned proofreading to Editage, a professional proofreading company.

Reviewer 2 Report

This manuscript covers an interesting and much needed research “The QPLEX™ Plus Assay Kit for Early Clinical Diagnosis of  Alzheimer's Disease”. Overall, the manuscript has innovative strengths, but also various weaknesses, as outlined below. I have suggestions below as to how the manuscript could be improved.

 The author should mention some relevant information in the introduction section from recently published “2023 Alzheimer's disease facts and figures” data (PMID: 36918389).

 Statistical analysis can be improved, It seems that study was conducted in different cohorts while in analysis cohort was not considered as variable.

The limitation section should be more explanatory as control participant in ApoE ε4 positive group is 8 which was not a large enough sample size for statistical significance analysis, but it is reported in limitation section.

The cutoff threshold sensitivity value of QPLEX™ Alz plus assay kit should be validate at least by two methods.

Also, author should mention the cutoff threshold sensitivity value of QPLEX™ for all four biomarkers LGALS3BP, Aβ40, ACE, and POSTN.

Author Response

Q1) The author should mention some relevant information in the introduction section from recently published “2023 Alzheimer's disease facts and figures” data (PMID: 36918389).

A1) Thank you for your comment. According to your recommendation, we added the following sentences to the introduction and discussion sections:

“As of 2023, the prevalence of Alzheimer's dementia among the older population (aged 65 years and older) in the United States is estimated to be approximately 6.7 million individuals.” (page 1, lines 42-44).

“Based on estimates for the year 2023, it is projected that approximately 8%–11% of the American population aged 65 years and older, corresponding to approximately 5–7 million older individuals, may exhibit MCI” (page 2, lines 57-59).

“AD neuropathological changes initially target specific brain regions associated with memory, language, and cognitive functions. Consequently, the prodromal symptoms primarily present as impairments in memory, language, and cognitive abilities.” (page 9, lines 265-268).

Q2) Statistical analysis can be improved, It seems that study was conducted in different cohorts while in analysis cohort was not considered as variable.

A2) We recruited participants from multiple clinical sites, but performed clinical diagnosis, cognition tests, and blood sampling under the same protocol under the same project.

Q3) The limitation section should be more explanatory as control participant in ApoE ε4 positive group is 8 which was not a large enough sample size for statistical significance analysis, but it is reported in limitation section.

A3) As per your advice, we added an explanation to the limitations section (page 10, lines 335-339).

Q4) The cutoff threshold sensitivity value of QPLEX™ Alz plus assay kit should be validate at least by two methods.

A4) The algorithm coefficients and cutoff value were validated in PMID: 33407839. The cutoff value was derived from ROC and was validated using a randomized sample selection method. The slightly different cutoff value is a corrected version through subsequent research after publication. However, since it is only a research result with additional samples within the same cohort, it was not published separately.

Q5) Also, author should mention the cutoff threshold sensitivity value of QPLEX™ for all four biomarkers LGALS3BP, Aβ40, ACE, and POSTN.

A5) Each individual biomarker did not show as much significance as in the QPLEX kit, so the AUC was not high and there was no clear value to choose as a cutoff. This is why we combined four biomarkers and algorithmized them. Only after increasing the discrimination by combining them, we were able to choose an appropriate cutoff. Even among participants with the same algorithm value, the quantitative values of each biomarker are often different.

The attached manuscript reflects all the comments and suggestions of the reviewers.
For the quality of English, we commissioned proofreading to Editage, a professional proofreading company.

Round 2

Reviewer 1 Report

I congratulate the authors for significantly improving the manuscript and following most of the recommendations. For the manuscript to be published, it would only need to include the results associated with education, high blood pressure, diabetes, hyperlipidemia, stroke, angina, thyroid, surgical history, cancer, family history, alcohol consumption, smoking, BMI, and anxiety.

Minor editing of English language required

Author Response

Thank you very much.
Your advice has been very helpful to us.

According to your recommendation, we added the following sentences to the introduction section: "Moreover, we checked the influence of factors such as education years, hypertension, diabetes, hyperlipidemia, stroke, angina, thyroid, surgical history, cancer, family history, drinking, smoking, body mass index (BMI), and anxiety. However, at least in our data, there were no significant differences between clinically separated groups (Supplementary Table S1), so we did not include them in the in-depth analysis." (page 3, lines 125-129).

The analysis results for education, hypertension, and so on were added as supplementary data.
